# Challenges and Approaches of Culturing the Unculturable Archaea

**DOI:** 10.3390/biology12121499

**Published:** 2023-12-07

**Authors:** Muhammad Rafiq, Noor Hassan, Maliha Rehman, Muhammad Hayat, Gullasht Nadeem, Farwa Hassan, Naveed Iqbal, Hazrat Ali, Sahib Zada, Yingqian Kang, Wasim Sajjad, Muhsin Jamal

**Affiliations:** 1Department of Microbiology, Faculty of Life Sciences and Informatics, Balochistan University of IT, Engineering and Management Sciences, Quetta 87100, Pakistan; 2FF Institute (Huzhou) Co., Ltd., Huzhou 313000, China; 3Industrial Biotechnology Division, National Institute for Biotechnology and Genetic Engineering-College, Pakistan Institute of Engineering and Applied Sciences, Islamabad 44000, Pakistan; 4State Key Laboratory of Microbial Technology, Institute of Microbial Technology, Shandong University, Jinan 266101, China; 5Department of Biotechnology, Faculty of Life Sciences and Informatics, Balochistan University of IT, Engineering and Management Sciences, Quetta 87100, Pakistan; 6The Department of Paediatrics and Child Health, Aga Khan University, Karachi 74800, Pakistan; 7Guangzhou Institute of Energy Conservation, Chinese Academy of Sciences, Guangzhou 510640, China; 8Key Laboratory of Environmental Pollution Monitoring and Disease Control, Ministry of Education of Guizhou, Guiyang 550025, China; 9Key Laboratory of Medical Microbiology and Parasitology, School of Basic Medical Sciences, Guizhou Medical University, Guiyang 550025, China; 10Department of Biological Sciences, National University of Medical Sciences, Rawalpindi 46000, Pakistan; 11Department of Microbiology, Abdul Wali Khan University, Garden Campus, Mardan 23200, Pakistan

**Keywords:** archaeal cultivation, archaeal diversity, culturing media, culturing methods

## Abstract

**Simple Summary:**

Living organisms are divided into three main domains: Domain bacteria and domain archaea have prokaryotic unicellular microbes, while domain eukarya have complex eukaryotic organisms. The two domains, eukarya and bacteria, are more extensively studied for the taxonomy, diversity, ecology, physiology, and clinical and biotechnological aspects compared to the archaea. The laboratory cultivation of archaea is more challenging than bacteria and eukaryotic microbes, making archaea less explored than bacteria and eukarya. The obstacles in the isolation of unculturable archaeal species arise from various known and unknown factors, demanding precise physicochemical and environmental conditions for their growth in the laboratory. Contrary to the earlier belief that archaea thrive only in extreme environments, recent reports suggest their ubiquitous presence, similar to bacteria but difficult to isolate. This review demonstrates the importance of archaeal culturing, challenges in cultivation, current approaches, and proposes future recommendations for successful isolation and investigations of the archaea.

**Abstract:**

Since Carl Woese’s discovery of archaea as a third domain of life, numerous archaeal species have been discovered, yet archaeal diversity is poorly characterized. Culturing archaea is complicated, but several queries about archaeal cell biology, evolution, physiology, and diversity need to be solved by culturing and culture-dependent techniques. Increasing interest in demand for innovative culturing methods has led to various technological and methodological advances. The current review explains frequent hurdles hindering uncultured archaea isolation and discusses features for more archaeal cultivation. This review also discusses successful strategies and available media for archaeal culturing, which might be helpful for future culturing practices.

## 1. Introduction

Archaea are recognized as an essential realm of life, occupying a central role in Carl Woese’s Tree of Life [1,2,3,4,5]. Archaeal signatures are evident and prevalent in every ecosystem around the globe, including hypersaline habitats [6,7], high temperatures [8,9], acidic environments [10,11,12], low temperatures [13,14,15], and typical environments like soils and oceans [16]. Archaea constitute approximately 20% of the total microbial communities of marine ecosystems [17,18,19,20]. Earlier, all the archaea were considered to belong to two primary phyla, the Euryarcheota and Crenarchaeota [7,21,22,23]. Over the past two decades, archaea have emerged into the research spotlight, garnering increased attention and undergoing comprehensive exploration across diverse habitats, ranging from extreme to conventional environments. Currently, an estimated number of archaeal phyla in the SILVA database based on ribosomal small unit gene sequencing is 30, containing 20,000 species [24,25]. At the same time, the taxonomic classification based on the genome at the Genome Taxonomy Database (GTDB) has 21 phyla with 7777 species [25,26,27]. Within the increased discovery of new species, the archaeal taxa are classified into three superphyla: Asgard, DPANN, and TACK. The superphyla DPANN and TACK are named based on the phyla present in them, such as TACK referring to the Thaumarchaeota, Aigarchaeota, Crenarchaeota, and Korarchaeota and DPANN to the Diapherotrites, Parvarchaeota, Aenigmarchaeota, Nanohaloarchaeota, and Nanoarchaeota [8,28]. In addition, the superphylum Asgard consists of phyla, e.g., Heimdallarchaeota, Helarcheota, Lokiarchaeota, Odinarchaeota, and Thorarchaeota [29,30,31,32,33]. However, one phylum, the Euryarcheota, is not categorized as a superphylum. These phyla and species contain culturable and nonculturable archaeal genomes reconstructed from metagenome assembly from different habitats [25,26]. The number of unculturable genomes is much higher than that of culturable archaea. Data regarding uncultivable archaeal predominance have been extended over the past decades [34,35,36] and widely reported for the critical ecological roles in diverse habitats [37].

Despite the strengths of metagenomic exploration, the isolation of archaea in the laboratory becomes invaluable for a comprehensive understanding of their distinct physiological and cellular properties, metabolism, nutritional requirements, genetic makeup, and biotechnological potential [38,39]. The isolation of the archaea is challenging because it is difficult to meet the requirements (physicochemical and nutritional) for the cultivation of archaea. Consequently, conventional methods like serial dilution, basic nutrient media, and standard plate methods for archaeal isolation prove less efficient in achieving increased archaeal isolates [40,41,42]. Owing to the limited understanding of archaeal growth requirements, providing optimum nutrients, salts, pH, temperature, and gasses required for the growth of archaea in the laboratory is arduous. How to achieve all the factors for archaeal isolation? The pursuit of optimal conditions for the isolation of archaea has sparked scientific inquiry into the intricate factors involved. These questions have inspired scientists to discover innovative methods for the cultivation of archaea [43].

Advancement in microbial genomics, metagenomics, and phylogenetic methodologies has significantly increased researcher’s knowledge regarding microbial diversity and their lifestyle [35,36]. Based on prior knowledge, more bacterial and fungal species are isolated, in contrast to archaea isolation, which is more challenging. In such cases, advanced genomic and metabolomic approaches prove very helpful in understanding the taxonomic and functional metabolic profiling of archaea within any environment. Subsequently, these analyses contribute to elucidating optimal culturomics conditions for the isolation of archaea through the reverse genomics approach. Microorganisms from these uncultured groups need to be cultured to explore the genome-based cell physiology predictions and accurately recognize their ecological role in their respective habitats. Their ecological role can be highlighted by discovering novel pathways and enzymatic reactions through experimental testing by culturing enrichment methods.

## 2. Significance of Archaea and Archaeal Culturomics

Domain archaea have great importance and play a vital role in ecological cycles. The role of archaea is not fully explored yet as most of them are not cultured in the laboratory. Still, archaea’s different functions and importance are studied in the environment. Archaea play roles in ecosystems by participating in biogeochemical processes such as anaerobic methane oxidation, organic carbon decay, nitrogen and carbon cycling ammonia oxidation, as well as having parasitic or symbiotic relationships with other species [44,45,46,47]. Archaea from the Euryarcheota group are known as methanogens and methanotrophs, hence playing an essential role in producing and reducing greenhouse gasses. They are the only organisms that anaerobically metabolize methane discharged from reservoirs of marine sediments at benthic zones. Therefore, these archaeal species are fundamental regulators of atmospheric methane [44]. Marine group-II and group-III archaea of Euryarcheota (also referred to as Poseidoniales and Pontarchaea) are efficient decomposers of organic carbon compounds in deeper marine habitats [47]. No cultured clade of group-III and -IV archaea is reported, prevalent in various marine niches [48].

Furthermore, TACK archaea represent anoxic and methanogenic representatives outside Euryarcheota [49,50]. Similarly, Bathyarchaeota [TACK archaea] are efficient universal and generous metabolizers underneath anoxic sediments. TACK archaea include some known lineages, i.e., *Bathyarchaeota* and *Verstraetearchaeota*, that are outside of the phylum Euryarcheota clade [49,50]. In addition, Phyla Odinarchaeota, Heimdallarchaeota, Thorarchaeota, Lokiarchaeota, and Helarcheota are mostly recovered from hydrothermal vents and marine sediments belonging to super phylum Asgard. Phylum Heimdallarchaeota is a central and essential target for culturing as it currently creates a very close lineage sister group to eukaryotes [32]. Super phylum DPANN has been reported from various habitats with a small genome and minimal metabolic abilities and is most likely dependent on other organisms in parasitic or symbiotic association [51]. Although this group of archaea is metabolically constrained, few have been reported utilizing organic compounds, such as carbon and lipids, via glycolysis, beta-oxidation, and other pathways [5,52].

Previously, the archaea were considered nonpathogenic organisms and were not reported for any clinical manifestation in humans, animals, or plants. However, the report of Hassani et al., 2022 recently opened a new window and aspect of archaeal investigations. They reported and recovered archaea from an oral sample linked to human periodontitis infections [42].

It is believed that the significance of archaea goes beyond our current understanding. The isolation of archaea is mainly attributed to the lack of knowledge and methods for isolating uncultured archaea. To better understand uncultured archaea, it is essential to enhance our knowledge to bring them from the natural environment into the culture in a laboratory [53,54]. Prioritizing cultivation is supposed to be most exciting as it adds more knowledge about a particularly ill-characterized group or improves our understanding of a particular process [55]. Culturing has prime importance for understanding several biological features for all single-cell and multicellular microbes from eukaryotes (protozoal group, fungi), prokaryotes (bacteria), and members of domain archaea.

## 3. Challenges in Archaeal Culturomics

Various cultivation methods have an excellent capacity for archaea isolation. Nonetheless, their accomplishment is not successful across the archaeal domain. Hypothetically, these techniques can be applied to isolate various archaeal taxa; however, their actual applicability might be more challenging in several cases. Archaeal isolation is more challenging than that of bacteria [56,57]. There is a surge in archaeal genome sequences. However, cultivable archaeal isolates are very limited [17,58]. Archaeal growth requires an energy supply of chemicals, nutrients, and adequate physiochemical conditions. Formulation of precise culture media to support archaeal growth is challenging because different archaeal species have diverse nutritional requirements.

Similarly, with a limited understanding of the archaeal optimal growth conditions, fewer archaea are retrieved on the plate count method to explore the archaeal diversity and actual role in the natural world [56,59,60]. Additionally, our understanding and knowledge about archaeal cultivation are limited by multiple factors; therefore, chances of archaeal cultivation in laboratories are minimal [56]. Some of the factors are discussed here.

### 3.1. Environmental Conditions/Abiotic Conditions

Unusual physiochemical conditions like pressure, redox condition, salinity, temperature, and pH are significant factors for archaeal cultivation. However, these determinants vary significantly in natural habitats at microscale distances. In nature, certain microorganisms subsidize the development of favorable environments for others, making it difficult to isolate such organisms that depend on these factors. Similarly, biotic and abiotic factors for archaeal growth in a laboratory setting differ greatly from the natural environments. These factors limit the cultivable archaea to only a few isolates [61,62]. Several strategies have been reported for archaea’s successful isolation and growth by coinciding the growth conditions with environmental conditions regarding temperature, pH, nutrients, pressure, and gasses. Notably, Brown and Kelly (1989) [63] reported a method to grow hyperthermophilic, anaerobic archaeon *Pyrococcus furiosus* by providing high temperature, anaerobic conditions, controlled pH, and nutrients under specific conditions. This involved the development of a control system in a bioreactor-type vessel that facilitated the growth of *Pyrococcus furiosus* in seawater (with composition detailed in the reference paper), yeast extract, and peptone. The successful growth was achieved by the addition of media content into the vessels, the temperature was set at 100 °C, anaerobic condition was achieved through N2 purging, and inlets and outlets were utilized for the addition and removal of nutrients and byproducts. Similarly, polyextremophilic archaea from the Shoor River in the Lut desert, Iran, experiencing high salinity, high irradiation, and periodic desiccation, were isolated by adopting specific strategies to provide maximum actual environmental conditions. Fifty-nine archaea were isolated using 23% of the salt-containing modified growth medium (MGM) media with successive incubation at 37 °C for two months. After the primary isolation, the archaea survive the desiccation for up to 8 weeks. [64]. A great deal of research must be pursued in extending cultivation approaches by optimizing the biotic and abiotic factors that facilitate the growth of uncultured archaea [19].

### 3.2. Suitable Nutritional Requirements (Concentration of Substrates)

Problems in archaeal cultivation have long been understood. The surge in genomic data has improved our understanding of microbial physiology since essential media components such as substrate, electron acceptor, and donor are essential for the growth of a particular microorganism. However, most microorganisms’ cultivation depends on supplementing numerous growth factors in the culture media, like nucleotides, amino acids, vitamins, and inorganic compounds in specific concentrations [65,66]. Similarly, several micronutrients such as nitrogen, metals, and sulfur must be present in trace amounts and should not exceed the maximum tolerated concentrations. Formulation of a particular media and supplementing desired organic and inorganic compounds are challenging tasks [67]. For example, archaeal growth can be halted if archaea are exposed to a high concentration of glucose, phosphate, or ammonia [65]. Isolating novel species from hypersaline environments would undoubtedly be aided by developing new unique culture media containing high salt concentration and growth conditions [68], and this phenomenon can be applied to isolate archaea from other habitats with specific geochemistry and other conditions. The substantial knowledge about the targeted archaea, its physiology, and the in situ conditions of its surrounding environment can facilitate the formulation of media and other physical requirements for growth in the laboratory. However, preference for species and strain-dependent substrates often fails in isolation when the specific medium is employed to distinguish closely related archaea species. Media formulation should be considered very wisely as closely related archaea, such as *Geoglobus acetivorans* and *Geoglobus ahangari*, could not grow in a single media [69,70,71]. Currently, the majority of archaea belong to uncultivable clades [72,73] and can be linked to syntropy, lack of knowledge, and difficulty of suitable media formulation [65,73]. The best mimic conditions (chemical and physical) of archaeal habitats enhanced the likelihood of uncultured archaea culturing [74].

Microorganisms in their natural habitats are adapted to limited substrate supply in oligotrophic conditions and other environmental circumstances [75]. Formulation of such media and conditions corresponds with the successful isolation of *Nitrosophaera gargensis*, *Nitrosophaera maritimus,* as well as a few methoxydotrophic and methanogens [76,77]. An increase in nutrient concentration, such as glucose, phosphate, and ammonia [65], can halt archaeal growth [75,76]. Several factors like sodium pyruvate, catalase, superoxide dismutase, and glycine betaine can negatively regulate archaeal growth if they exceed the optimum concentration and can be eluded while isolating such a kind of archaea [74].

Similarly, varying compositions and appropriate concentrations of energy sources such as lignin, oleic acid, phenol, casein, and cellulose were tested for culturing of Bathyarchaeota from estuarine sediments, though only lignin was an effective carbon source for the growth of archaea [17,78]. The factors behind using lignin as a source of energy by the specific group of archaea recovered from the marine sediment have multiple copies of genes coding for lignin-degrading enzymes as well as its transportation and utilization as a carbon and energy source. The evolutionary mechanisms over time in lignin-rich extreme habitats enable Bathyarchaechaeota to gain functions in terms of genes for lignin utilization. Therefore, the chemical nature of the habitat is also helpful in the formulation of media for the isolation of uncultured archaea.

### 3.3. Consortium and Co-Culturing

In a natural environment, archaea are living in community structures. Several archaea with smaller genomes depend on other archaea for their growth and cannot be isolated as a pure culture until specific nutrients are provided, which they achieve from co-culturing [56]. Alternatively, interactions among archaeal communities cannot be established at the laboratory level, leading to archaeal cultivation failure. Failure to achieve archaeal cultivation could be linked to the inhibition of signaling molecules, unavailability of growth factors, and metabolic substances produced by another organism in situ [79]. However, if microbial communities were grown in a media with some inducers and signaling molecules like cAMP and Homoserine Lactones, the growth rate would be enhanced and reduce the doubling time. Some microbes produce these molecules in a co-culture, which others use for their growth [80].

### 3.4. Inappropriate Incubation Time

Inappropriate incubation time and unavailability of sensitive detection methods can also lead to failure in archaea cultivation [62,77]. The low abundance and slow growth of archaea in the sample resulted in an overgrowth of undesired microorganisms, leading to the inability to isolate particular archaea [81]. Incubation failures could also be due to improper transportation of samples from the natural environment [82]. Consequently, patience, precautions, and accurate confirmation are required for any cultivation [74,83]. The choice of incubation period limits archaea’s number, nature, and diversity from varying habitats [74].

### 3.5. Low Abundance, Presence of Persisters, and Dead Cells

In natural settings, most prokaryotes are present as complex microbial communities at low abundance, though they could still have considerable impacts on several processes [84]. Low abundance could rise, linked with a slower metabolic rate than other microorganisms in the same environment. Such microorganisms are quickly outcompeted by faster-growing organisms; even if these microorganisms are initially abundant, their comparative richness will decrease when they are cultured together. To successfully isolate such rare microorganisms, finding habitats in which these microorganisms are predominant could help additional enrichment efforts. Similarly, many microorganisms (Archaea) are currently unculturable and possibly dominated by persisters [85], which are non-dividing and dormant, representing a default stage of prokaryotic life [86].

Persister cells are commonly present in the population of bacteria. However, they can also be present in archaea [87]. The revival of dormancy signifies a crucial obstacle in the cultivation efforts of microorganisms. Considerable data regarding microbial dormancy are present, though probable mechanisms reinforce that it is unknown how shifts between microbial dormancy and active state occur. Some microbes in natural habitats also remain dormant; for successful isolates, revoking them from dormancy to an active state is crucial. According to Esptein’s mathematical model, dormancy and revival patterns are random, complicating the isolation of targeted dormant groups [88]. The isolation can be increased if there is a possible mechanism to understand and regulate quiescent microorganisms. Dead cells in natural settings could also lead to failure of microbial cultivation [65] as in total environmental DNA, where about 40% is from compromised or dead cells in temperate soil [55]. These factors create difficulty in both bacterial and archaeal culturing in laboratory media. Interaction is vital for growth; one community may produce essential products that benefit archaea’s development (Figure 1).

### 3.6. Small Genome Size of Archaea (Symbiotic Association)

Some groups of archaea contain small cell sizes and even smaller genomes, due to which they lack some essential metabolic genes and pathways. However, some of these pathways are needed for critical features and cell growth, making them very difficult to cultivate [28,52]. Therefore, members of Diapherotites, Parvarchaeota, Aenigmarchaeota, Nanoarchaeota, and Nanohaloarchaeota (phyla of DPANN archaea) are reported from several natural habitats like marine sediments, hot springs, acid mine drainage, and different water sources [28,52,79,89,90,91,92]. Most of these archaea are non-cultivable; however, a few representative clades suggest that DPANN archaea are usually found in symbiotic association with other archaea, bacterial, or fungal species. Even parasitism is also observed in some cases [35]. This requirement provides further obstacles for DPANN archaeal cultivation as suitable requirements must be fulfilled to satisfy both archaeal partners and appropriate knowledge of the relation between partners.

### 3.7. Incorrect Genome Annotation

Data about archaeal genome sequences are scarce due to the fewer number of cultivable species of archaea, which depicts the unavailability of more genome sequences for annotation and extraction of proper metabolic pathways [93]. Therefore, some archaeal genes may have been wrongly annotated or linked with unknown and hypothetical proteins, indicating a genomic lack of information for archaeal cultivation. [17,56,93]. The reports denoted that genomes of archaea are annotated with hypothetical proteins up to 30% and as much as up to 80% in some instances and consist of dark genomic matter [93]. Therefore, based on the exact genome annotation, the functional profile and biosynthetic pathways can be mined for specific archaeal species. Furthermore, a media can be formulated through reverse genomics, providing all the necessary nutrients aligned with the metabolic potential of the targeted metagenomically established uncultured archaea. In the case of actual annotation, it will also be more helpful to avoid adding such chemicals into the media, which causes hindrance to the archaeal growth. Similarly, numerous microorganisms may grow and utilize a variety of substrates, which could be hypothesized from genomic data. For instance, genomic information revealed that Asgard archaea and Bathyarchaeota could use complex organic substances like hydrocarbons, alcohol, fatty acids, peptides, amino acids, and carbohydrates [45,94]. Challenges in archaeal cultivation are given in (Figure 2).

## 4. Current Status of Culturomics for Archaea

The exact number of archaeal species is expected to be much higher than that cultured in the laboratory. According to 16S rDNA sequences submitted to different databases, the estimated number of bacterial and archaeal genera and species are about 60,000 and 400,000, respectively [95]. However, real bacterial and archaeal species on the planet probably surpass this number [96,97,98].

Microbial cultivation started in the mid-19th century. Most current cultivation attempts depend on several initial concepts introduced more than ten decades ago [99]. Different strategies can be used to isolate a particular microorganism, and the isolation depends on phenotypic, genotypic, and physiological characteristics that vary for each organism. Among prokaryotes, earlier studied microorganisms were archaea, but fundamental differences between bacteria and archaea were not known until Carl Woese’s discovery in 1977 [1]. Major metabolic activities of different archaea were evident even before discovering the microscope, for instance, the production of marsh gas (methane) by anaerobic methanogenic archaea.

Similarly, hyper-thermophilic archaeal species found in hydrothermal vents were reported in the late 20th century. In 1868, microorganisms’ role in methanogenesis was discovered, but linking it with archaea took another century [69]. Pure methanogenic archaeal species, *Methanosarcina barkeri*, *Methanococcus vannielii*, and *Methanobacterium formicicum*, were isolated from a complex microbial community in the mid-twentieth century [41].

Furthermore, halophilic archaea are the second most abundant archaeal domain, consisting of initial representatives, e.g., *Halobacterium salinarum*, *Halobacterium cutirubrum,* and *Halobacterium halobium*. In 1960, Sehgal and Gibbons [100] reported the outcome of metal ions on halophilic cultures of *Halobacterium salinarium*, *Halobacterium cutrirubrum*, and *Halobacterium halobium*. Furthermore, a chemically defined media for halobacterium strain was developed by [101]. Oren (1983) [102] reported a novel species of Euryarcheota, *Halobacterium sodomense* from the Dead Sea. Halophilic archaea, *Halobacterium* and *Halococcus*, were cultured aerobically for determination of polyamines quality and quantity [103]. Moreover, *Halobacterium volcanii*, an extremely halophilic member of class archaea from phylum Euryarcheota, was identified as a native gene transfer mechanism [104]. Oren noted extreme halophilic characteristics of the family haloarchaea. A novel extreme halophilic archaeon specie *Halobaculum gomorrense* was reported and characterized from the Dead Sea [105]. Zillig et al. (1990) [106] identified several thermophilic archaeal strains from microbial communities after H. Jannasch and M. J. Mott’s discovery from deep-sea hydrothermal vents. Hence, thermophilic archaea became well known through culture-independent and culture-depended techniques that comprise *Sulfolobus* and *Thermoplasma*. Genome sequencing, combined with metagenomics and environmental studies, greatly enhances knowledge of domain archaea.

In addition, a new genus of sulfur-oxidizing archaea, *Sulfolobus*, was isolated and characterized from acidic hot springs and soil [107]. Therefore, Brock and his colleagues (1972) modified basal salt media formulated by Allen [108] to cultivate and characterize *Sulfolobus* archaeon species. Kester et al. (1967) [109] prepared an artificial seawater-based culture media for the isolation of hyperthermophilic archaea (*Pyrococcus furiosus*) belonging to phylum Euryarcheota. Another novel thermophilic genus, Pyrobaculum, was reported by Huber et al. (1987) [110] in Crenarchaeota. A cultivation technique for isolating hyperthermophilic archaea *Pyrococcus furiosus* was developed in 1989 [63]. Recently, diverse genomic sequences from culture-independent methods have led to more kingdoms to domain archaea like Nanoarchaeota and Korarchaeota.

The co-culture method, in combination with enrichment and the single-cell sorting approach, was used for isolation of *Nanohaloarchaeum antarcticus*, co-cultured with Euryarcheota strain *Halorubrum lacusprofundi* determined by using fluorescence in situ hybridization (FISH) technique [111]. Some archaea are very slow growers, and some extraordinary approaches are required for successful isolation. A notable member of Asgard archaea *Candidatus Prometheoarchaeum syntrophicum* was successfully grown through co-cultivation using a conventional enrichment in a bioreactor for a long-run experiment of up to 12 years. Coping with a new cultivation environment and slow metabolic rate takes a longer time for successful growth [112]. Similarly, the first ethane-oxidizing closely related archaeal genera, *Candidatus Ethanoperedens thermophilum* reported by Hahn et al. (2020) [113] and *Candidatus Argoarchaeum ethanivorans* reported by Chen et al. (2019), [114] were isolated by specific enrichment method in a cross-feeding (syntrophic) relationship with bacteria capable of sulfate-reduction. Several methods used for successful isolation of archaea are summarized in (Figure 3).

In an advanced reverse genomic approach, Cross et al. (2019) [115] introduced a method that leverages genomic data-driven antigens on the targeted bacterial cell surface through reverse genomics to isolate oral bacteria successfully. This multidisciplinary, innovative approach also holds promise for isolating archaea from diverse habitats. By applying the reverse genomic approach, they isolated Saccharibacteria (TM7) and the less abundant group SR1/Absconditabacteria, marking a significant milestone in laboratory isolation through reverse genomics. The approach involved mining metagenome-assembled genomes (MAGs) to identify and synthesize Ags (Epitope) on the outer membrane unique to a specific targeted uncultured microbial group. This antigen (Ag) was employed to generate specific antibodies labelled with fluorescence, capable of binding with the epitope on the target cell’s surface in the microbial communities. The fluorescently labelled Abs are then introduced to the target samples, where the Abs binds to the target cells. The fluorescently labelled targeted cells were recovered through fluorescence-activated cell sorting (FACS) and used as an inoculum. The labelled cells were grown in the culture media [115]. A notable aspect of Cross’s experiment is the effective isolation of target microbes using labelled antibodies (Abs) while maintaining the viability of the cells. This approach is particularly valuable when other requirements for culturing are known for any specific strain. Likewise, the genomic data obtained from various metagenome-assembled genomes (MAGs) of archaea could be utilized to successfully isolate strains from different uncultured habitats [115,116].

## 5. Culturomics Strategies for Archaea

Research on archaea can be vastly enhanced by culturing the uncultured clads, i.e., enabling them to be alive under laboratory conditions by reproducing population. Despite enormous data on transcriptome and genome, obstacles in archaeal cultivation remain, including unidentified required nutrients (qualitatively and quantitatively), physicochemical parameters, and massive deviations of these requirements even between two close species. Therefore, developments in culture-dependent approaches for archaea are an essential need, and advancement in culturing should be a study hotspot [17]. In combination with some innovative cultivation methods, new strategies are needed for breakthroughs in the isolation and culturing of archaea [17].

### 5.1. Simulation Strategies of Natural Habitat (High-Throughput Culturing)

The primary reason for the uncultivability of the microbes living and thriving in the natural habitats is the differences in terms of conditions (physical, chemical, environmental, molecular, and biological) of the natural environment and lab settings. To bridge this gap, diverse high-throughput culturing approaches are employed to cultivate a broader spectrum of microbes in the laboratory, simulating the complexities of their natural habitats. Among the most significant innovations are the development of a diffusion chamber [117] and the isolation chip (Ichip) [118], both designed to mimic the conditions found in natural habitats.

In the diffusion chamber, microbes are allowed to thrive in conditions analogous to their natural environment. The targeted inoculum or samples are sandwiched between two semipermeable membranes and incubated in the natural environment, allowing the movement of the nutrients and chemical signaling molecules. This method provides a favorable natural environment for the growth of microorganisms from diverse habitats. Similarly, in the Ichip, a device with numerous tiny chambers, each chamber provides environmental conditions for a single cell from the samples. In the Ichip, the chambers are separated by small-size selective permeable membranes (pore size 0.03-µm), allowing the exchange of small molecules and preventing cell escape from the chamber. The Ichip allows the exchange of growth factors and other nutrients and is incubated in the natural environment to attain all the environmental conditions required to increase growth by many folds. Based on the foundations of the diffusion chamber and the Ichip, many habitats are explored for the isolation of microbes with encouraging results, e.g., freshwater [117,119], activated sludge saline soda lakes [120], hot springs [119], seawater [118], and other environments [121]. The key considerations in these high-throughput culturing approaches involve optimizing the simulation of the physicochemical and biological parameters of the environment to the highest degree. These include ensuring the availability of essential chemicals, exploring satellite and co-culturing growth, minimizing physical variations, and providing the necessary growth factors and metabolites.

### 5.2. Co-Culturing

The co-culture allows the microbes to communicate with each other effectively compared to the pure culture. Many archaea have small genome sizes, lacking crucial genes responsible for critical metabolic pathways. These archaea can only grow if suitable fastidious media are provided or if it is growing in the surroundings of another archaea capable of producing the required metabolite for the growth. This phenomenon is termed satellite growth. The phenomenon of dependency may be one-way or bidirectional. In instances like the growth of methanotrophic and methanogens, each provides an environment for the growth of the other. The methanotrophic archaea utilize oxygen and provide an anaerobic condition for methanogen, which produces CH_4_ utilized by the methanotrophs for the carbon source [61]. Many successful examples of archaea isolated through the co-culture approach include recent clinical pathogenic archaea from dental samples.

Similarly, the Asgard archaea *Candidatus Lokiarchaeum ossiferum* was isolated through the culturing enrichment method [122]. Rodrigues et al., 2023 also isolated members of Asgard archaea using water (sterilized through filtering) from the site and supplemented with different nutrients and antibiotics [112]. Before selecting the sampling site, the abundance of Lokiarchaea in the samples was estimated through amplicon sequences to ensure their presence in the natural environment. The growth was periodically observed with the help of qPCR, and it was found that the best media for the enrichment and growth of Lokiarchaeata is minimized lokiarchaeal medium (MLM) with casein hydrolysate and filtered sterile water of the sampling site. Through this dual approach of co-culturing and enrichment, the abundance of Lokiarchaeon (*Candidatus Lokiarchaeum ossiferum*) was achieved up to 79%, followed by 10% of *Desulfovibrio genus* and 6% of Methanogenium. Other techniques like FISH, low-magnification 2D cryo-electron micrographs, cryo-tomograms, and rRNA structure are used to identify and confirm the isolated archaea through enriched co-culturing [122]. Similarly, the co-culturing of archaea with other archaea and bacteria is helpful for the initial isolation of the strains that cannot grow independently [17,42,49].

### 5.3. Single-Cell Sorting

The single-cell cultivation approach is an effective method for isolating eukaryotic cells and some bacteria. Different tools are used to select and collect a single microbial cell and transfer to the culture media. A thin hollow device like a microneedle or tube linked to a pressure device under the microscope picks up a single cell from a mixed culture [49,123]. Another approach of microencapsulation can also be used to isolate archaea in pure culture [124]. Microencapsulation is another single-cell sorting approach similar to the emulsion PCR technique of microdroplet formation in a gel. The mixed communities of microbial populations are suspended in gel microdroplets, and diffusion of low nutrients enhances the massively parallel growth in droplets before sorting and transfer to media [49,124].

Similarly, the infra laser beams (optical tweezers) are used to trap and collect single cells into a sterile syringe. This method is used to isolate hyperthermophilic small-size *Nanoarchaeum equitans archaea* [79,125]. The single-cell isolation method is only successful if the archaea grow on the media after isolation from the mixed community.

### 5.4. Culture Media for Halophilic Archaea

Growth of halophilic archaea required 1.7 to 2.5 M sodium chloride (NaCl) salt in media. However, most halophilic can grow at higher NaCl salt concentrations like 5.5 M. Haloarchaea live and thrive in higher salty natural habitats such as brine lakes, dead sea water, and marine solar salter [126,127,128]. Notably, highly halophilic archaea are usually found in brackish water with a high concentration of sodium chloride. Unusual properties of these archaea, including their need for high NaCl for optimum growth, limited ability to use carbohydrates, and need for proteins from external sources, have contributed to the conclusion that a well-developed enzyme complex and structural proteins responsible for the digestion of amino acids exist [129,130]. However, though many of the culturable isolates belong to the halophilic group, it is still challenging to culture the significant number of unculture halophilic archaea in lab media because they require extra vigilance compared to other prokaryotes. Different media for the cultivation of halophilic archaea are given in Appendix A.

### 5.5. Culture Media for Methanogenic Archaea

Methanogenic archaea are the only known life forms producing methane as a metabolic byproduct [17]. Methanogenic archaeal groups are fastidious in nature, and cultivation media should be supplemented with carbon dioxide and hydrogen sources [131]. Regular media are insufficient for the growth of methanogenic archaea as they need media that meet specific requirements [132]. Different media compositions are given in Appendix A for cultivating methanogenic archaea.

### 5.6. Culture Media for Thermophilic and Hyperthermophilic Archaea

Thermophilic archaea grow at a temperature above 80 °C, while hyperthermophiles at a temperature above l00 °C. Anaerobic conditions, gases such as methane and hydrogen sulfide, and high levels of inorganic salts are properties of their natural environment [133]. Culture enrichment for isolating thermophilic and hyperthermophilic archaea requires particular nutrients and trace elements [134], as described in Appendix A.

### 5.7. Culture Media for Mesophilic Archaea

Mesophilic archaea need fewer trace elements, including zinc, iron, cobalt, and nickel, and grow more easily than other archaea [135]. Crenarchaeotes belong to mesophilic archaea and can be grown at standard mesophilic range temperatures [136]. Mesophilic crenarchaeotes have ecological importance as they are significant in geochemical cycles. Crenarchaeotes are collected from the soil but are prevalent in oceans [137]. Growth media for mesophilic archaea are given in Appendix A.

### 5.8. Culture Media for Acidophiles Archaea

Acidophilic archaea survive in acidic habitats [138]. For example, members of *Sulfolobus* were isolated from the acidic habitat of Yellowstone National Park [107] by using a medium called Brock medium which contains both yeast extract and sulfur to isolate heterotrophic and autotrophic acidophiles. Media formulation for acidophiles is given in Appendix A.

### 5.9. Proposed Basal Media for Archaeal Cultivation

Cultivation of archaea is exciting but challenging. However, numerous nutrients and trace elements are essential for archaeal growth in their natural habitat. Formulation of specific media is required for archaeal growth in laboratories. However, when applying conventional cultivation approaches, only a small number of microbial communities of natural habitats can be isolated, while 99.9% of microorganisms are yet to be cultivated [37,56]. More than one approach should be adopted to mimic the natural habitats as closely as possible, in all terms of nutritional, physical, molecular, and biological processes, to successfully isolate the maximum number of archaea. The growth of archaea can also be enhanced by inhibiting the growth of other competitive and fast-growing microbes like bacteria and fungi by providing antibiotics like kanamycin and streptomycin [135,139] and antifungals like nystatin and azoxystrobin [140] in the culture media.

Most archaea dominate in extreme environmental conditions, e.g., extreme temperature, stress, and salinity. Yeast, peptone, NaHCO_3_, KCl, NaCl, MgCl_2_·6H_2_O, CaCl_2_·2H_2_O, KH_2_PO_4_, and NH_4_Cl can be used together as a basal medium. The majority of media with components have been reported for successful archaeal cultivation. Yeast extract and peptone are the prime carbon and nitrogen sources for microbial growth. In addition, vitamin B complex, sulfur, and trace nutrients are essential growth factors necessary for microbial cultivation. Kester et al. (1967) reported sodium bicarbonate [NaHCO_3_], which plays a significant role in survival in harsh environmental conditions as Na maintains cell integrity. Further, potassium chloride also plays a role in maintaining cell integrity [109].

Halophilic archaea sustain the balance between pulling out dissolved salt inside of cells. Salt-tolerant enzymes need K^+^ for proper protein folding and stability to manage high intracellular concentrations of K^+^ [141]. Yadav and his team also revealed that K concentration is vital for phosphate solubilization [142]. Archaea require NaCl between 2–5 M, but a minimum of 0.5 M NaCl is required for their growth [102]. Magnesium chloride [MgCl_2_·6H_2_O] and calcium chloride are also necessary for various cellular activities. KH_2_PO_4_ as a phosphorus and potassium source and ammonium chloride as a nitrogen source are essential components of all available media used to cultivate archaea. Based on current knowledge of archaeal culture media, the above-suggested ingredients are essential for all types of archaeal growth, but the other related factors are mentioned in the problems and limitations section.

## 6. Conclusions

Current methods and procedures employed in the cultivation of archaea do not signify the fraction of domain archaea members in the biosphere. Maturation and advancement of culturing technologies should be prioritized for the investigation of the presently underexplored archaeal domain of life. We have reviewed significant constraints that prevent the successful cultivation of archaea inhabiting varying habitats around the globe. Present available media components and constraints factors are considered in this review, representing avenues for future revolution and successful culturing practices. Based on available media, we suggested basal or general-purpose medium for culturing archaea. Profound genomic, metagenomic, and co-culturing studies might provide knowledge about the requirements of many archaeal groups and facilitate the formulation of new media for uncultured archaea.

## Figures and Tables

**Figure 1 biology-12-01499-f001:**
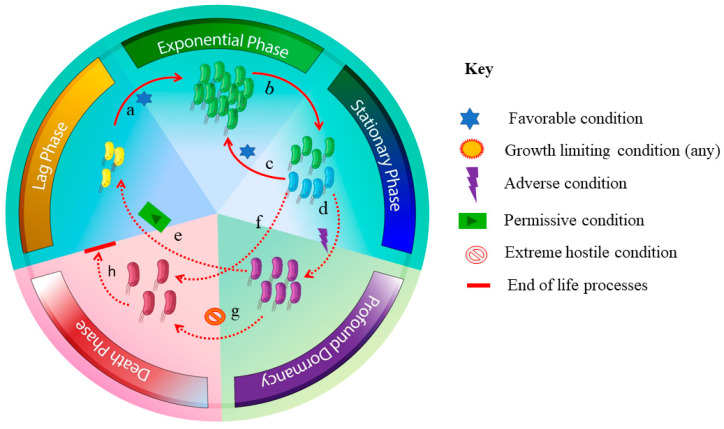
**Graphical representation of the archaeal transitions in diverse physiological conditions**. The diagram illustrates the **Lag phase**: the initial stage when archaea enter a new habitat, a stage where archaea adjust to their environment, access nutrients availability, and prepare for cell division. **Exponential phase**: a phase in which archaea replicate and divide actively, having a favorable condition. In this phase, the archaea are metabolically active, preparing cellular components and proteins. **Stationary phase:** growth slows down, and the number of active cells becomes equal to dead cells with a stable population size due to growth-limiting conditions leading to these conditions. However, when the conditions are favorable, the cells may transition to the exponential phase. **Profound dormancy:** In this phase, the archaea have limited growth and metabolism. They may also form a protective structure like cysts or spores, which make them survive in conditions that might be lethal to other organisms. **Death Phase:** In this stage, environmental conditions become inhospitable, leading to a decline in the archaeal population. These factors include nutrient deficiency and the build-up of toxic chemicals, due to which the population may die off eventually.

**Figure 2 biology-12-01499-f002:**
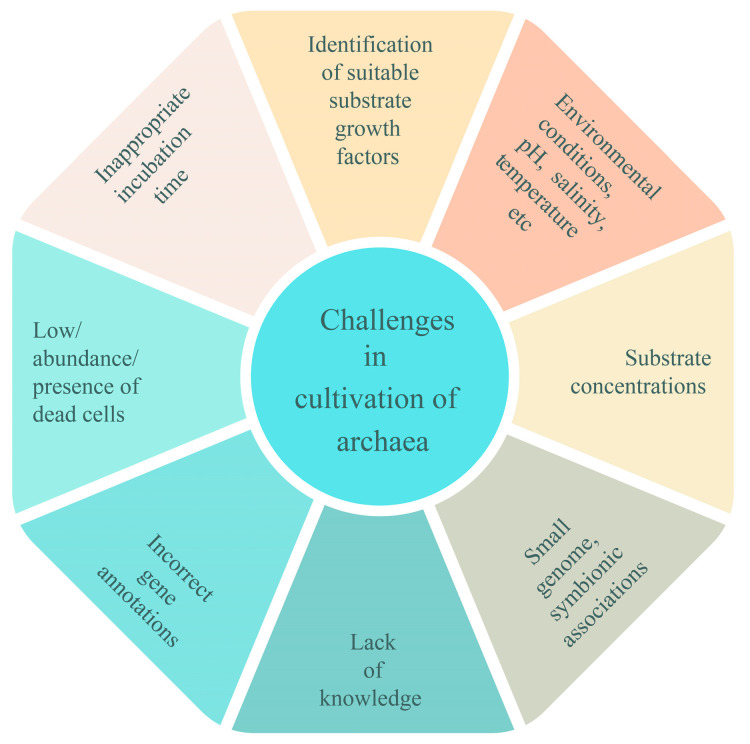
This figure represents various challenges for the cultivation of archaea.

**Figure 3 biology-12-01499-f003:**
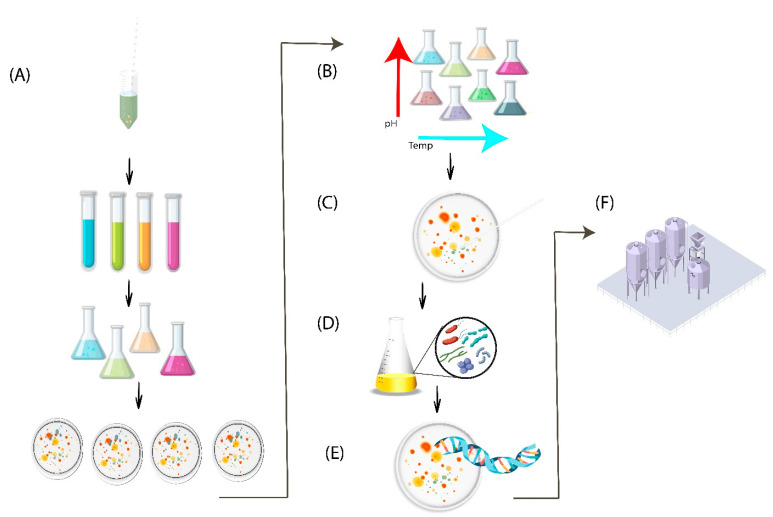
Methods for successful pure culture isolation of archaea. (**A**) Enrichment and screening of different media for archaeal isolation. (**B**) Various physiochemical conditions. (**C**) Colony picking from mixed culture. (**D**) Co-culturing or community culturing. (**E**) Using toxic substances or inhibitors and co-culturing coupled with fluorescence in situ hybridization. (**F**) Enrichment culturing combined with novel bioreactor.

## Data Availability

Data are contained within the article.

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
