# Peer review of "Challenges and Approaches of Culturing the Unculturable Archaea"

_biology, 2023, doi:10.3390/biology12121499_

Round 1

Reviewer 1 Report

Comments and Suggestions for Authors

The first impression is positive as Authors strongly relate to historical perspectives, rightly starting with Carl Woese's monumental discovery of the third domain of life. However, Authors should go beyond this perspective and highlight the latest developments in the field reported in recent research papers/review articles.

Please correct the description of Fig. 1. As stated in the text (lanes 71-72), cultivable Archaea belong to two major groups: Euryarcheota and Crenarcheota. However, Fig. 1 shows something contrary, indicating that Crenarcheota, Euryarcheota, and Thaumarcheota are unculturable. Theumarcheota species were successfully grown, as reported by Berg et al., 2015 (Front. Microbiol. 5:786) and Tourna et al., 2011 (PNAS 108:8420-8425).

In the current status of culturomics for archaea (section 5, starting with line 356), please give the overview of a strategy aimed at isolating and cultivating microbes by reverse genomics. This topic was nicely nailed in the report on human oral Saccharibacteria (Cross et al., 2019, Nat. Biotech. 37:1314-1321). This strategy is addressed in a review article by Liu et al., 2022 (Microbiome 10:76, Opportunities and challenges of using metagenomic data to bring uncultured microbes into cultivation).

While describing Culturomics strategies for Archaea, please mention the use of additives that can increase the cultivation efficiency of some Archaea, namely, antibiotics such as kanamycin and streptomycin that inhibit bacterial growth. Park et al., 2010 (AEM 76:7575-7587) and Simon et al., 2005) reported this kind of cultivation strategy (AEM 71:4751-4760). Also, the addition of fungal inhibitors, such as nystatin or azoxystrobin, would be beneficial in Archaea culturomics (Taylor et al., 2017, ISME J. 11:896, and Taylor et al., 2010, AEM 76:7691-7698).

In addition, While overviewing culturomics strategies for archaea, please do not focus only on culture media (section 5) but describe innovative cultivation techniques such as (i) co-culture, (ii) single-cell isolation, (iii) high-throughput culturing, (iv) reconstructive bioreactors, diffusion chambers, and iChips.

Comments on the Quality of English Language

The paper is written in understandable English

Author Response

We are thankful for the reviewers comments for the improvement of the article. We have made changes according to reviewers comment. 

Reviewer 2 Report

Comments and Suggestions for Authors

Reviewer’s comments:

The title initially intrigued me, but I found that the content in the manuscript did not adequately align with the title's promise. The manuscript requires a substantial rewrite to provide more comprehensive explanations and elaborations. The manuscript contains numerous paraphrasing and English language errors, some of which have been addressed in the revised version. However, it is essential to ensure that the entire manuscript is thoroughly reviewed for language and grammar issues. It appears that the manuscript may have been hastily assembled, with sections resembling a minimal publishable unit rather than a cohesive, well-elaborated review. I encourage the authors to invest more effort in enhancing the readability and enjoyability of the manuscript for readers. I insist on a major revision and addressing of the below edits and queries.

Question 1: How is this review work different from Sun paper of 2019: https://doi.org/10.1007/s00248-019-01422-7

Question 2: Has such a review been recently available in MDPI and/or other sources? If yes, what is new to read for us?

A few corrections and changes have been outlined here (REST IN THE DOCUMENT ATTACHED).

1.      Full address missing for affiliation 4.

2.      Introduction: I guess the context has not been staged properly. We need to discuss first about “why Archae are so very interesting to investigate”, then come up with the challenges and solutions for their culturing.

3.      The intro in real starts from here: line 68.

4.      Line 78: Enlist or discuss some – this is what catches the attention of laymen to read ahead and hold their interest.

5.      Line 92: This can go in the figure 1 caption better.

6.      Figure 1 labels: Spell check culturable and the other? CulturAble? Without an E there.

7.      Line 105 redundancy: You already said this before.

8.      106: How again, be specific. Cut short the generalities, please.

9.      Line 127: Why all these phyla can’t be put in Figure 1 to make it a more comprehensive infographic?

10.  139: Justifiable clarity please to the sentence?

11.  142: This counters your excellent capacity statement for isolation.

12.  148: You could have used MS-word spell and grammar check and the freely available Grammarly for the English language at least. These require language editing services otherwise.

13.  155: Discuss at least some successful trials with the biotic factors, otherwise it again stays as general statements.

14.  The most of the body text: Already discussed above. Please cite at right positions. See the statement, where, and why you have put citations. Please read and collate more data.

15.  196: Linguistics, please. Get good English language assistance, as none of the authors are native speakers. MDPI-language editing is highly recommended.

16.  196: You did not talk about the diversity of archea with genome size before.

17.  204: Discuss this cited work in brief if it shows what you wish to interpret.

18.  234: Clarify this please.

19.  263: Discuss some instances from these cited works and other more please to convince the reader: or Why should they just be directed to these papers to read even a glimpse of what data annotations were mistakenly affected in some previous works? Read articles in full and not just abstracts to get the essence of what these cited works have in them.

20.  263: Clarity please. What highlights substantial knowledge. What substantial knowledge? You just said the genome annotation is scarce. Then? Read and revise the manuscript for connectivity with and as a reader.

21.  266: What is this doing in this heading? Genome annotations?

22.  275: Yes, but they have not been discussed justifiably with proper examples. 

23.  421: We, please don’t use dialogues. Write opinions as descriptions.

24.  445: Rephrase for clarity.

25: The abstract does not provide a comprehensive overview of all the challenges and strategies mentioned in the title and what’s there as a glimpse of body text.

Regards,

The other reviewer. 

Comments on the Quality of English Language

I recommend thorough editing of English language. Use MDPI service preferably or a native English scientific editor. 

Author Response

We are thankful for taking the time to review this manuscript. Please find the detailed responses below and the corresponding revisions/corrections highlighted/in track changes in the re-submitted files

Reviewer 3 Report

Comments and Suggestions for Authors

Challenges and Culturomics strategies for Archaea: An Overview

Muhammad Rafiq, Noor Hassan, Maliha Rehman, Muhammad Hayat, Gullasht Nadeem, Farwa Hassan, Naveed Iqbal, Hazrat Ali, Sahib Zada, Ying-Qian Kang, Wasim Sajjad

General comments.

This paper was submitted as a review of the "challenges and culturomics strategies for Archaea. Let's begin with the title. Challenges of what? It is not clear what the challenges are from the title. The second phrase, "culturomics strategies" is also problematic. Why are both culturomics and strategies plural? Perhaps is should be culturomic strategies. What about "New strategies for culturing the unculturable Archaea". Or "The challenges of culturing the unculturable Archaea". These options are less confusing to me. The issue with the title repeats itself throughout the paper where poor grammar and imprecise use of language make reading difficult. Given the extent of this problem, only a complete rewrite paying critical attention to logical flow, proper sentence structure and precise use of words can bring the paper to a publishable level. Another fatal flaw is the level of repetition in the text. It is extensive and greatly detracts from a good read. This is an easy place to begin the tightening of the text. 

I also feel that the organization can be improved. The second heading titled "Significance Of Archaeal Culturomics" could easily be rolled into the Intro. The third heading titled "Challenges In Archaeal Culturomics" is logical but the information contained within this section is fairly superficial. For example the authors state "Several Archaea with smaller genomes depend on other Archaea for their growth and cannot be isolated as a pure culture (Alain and Querellou, 2009)." It may be that we simply don't know the essential nutrient supplied by the co-culture and by knowing this the strain would be culturable. The authors should take a more critical view of the literature and make specific suggestions regarding improved cultivation approaches rather than simply list several observation from the literature. To critically review all media used for the cultivation of Archaea would be a valuable contribution and then to link media components with successful cultivation of a sequenced Archaea would excite the reader. A series of tables listing media components culled from the literature is not terribly helpful. 

The topic of the review is worthy of a critical review and I think the authors are capable of providing a valuable resource for the community.

Specific comments.

L32   (Cross-Out) delete "to represent avenues"

L32  Cross-Out) delete "revolution and successful"

L41  to cultivate

L42-43  This sentence makes no sense. Needs to be recast.

L47  various significant what??

L36  The intro should be rewritten to be more concise and terse. As it stands it is highly repetitious with repeats stringing throughout the manuscript.

L48 substitute confirming for examining.

L49  and to accurately...

L52  This is not clear. First you are speaking of discovering novel pathways then you state that the pathways were not "earlier detectable" by genomic data" suggesting that the pathways are now known. This is confusing.)

L54  I think this is a given. Hardly needs stating.

L55  substitute "are required" for "could be used"

L60 wordy is "uncultivable nature under laboratory conditions"

L64  who is they?

L66 eukaryotes, bacteria and archaea are not functions. Language must be cleaned up considerably.

L72-73  repetitious

L80-82. again this is quite repetitious

L87  Grammar "considered as belong to" ???

L89  Grammar  "has enabled the description""...)

L95  Grammar Cross-Out) "consistes of phyla Heim... etc"

L106-107  Because the archaea are "one complete domain" does not indicate that they have strong roles in ecological cycles.

L106-107  Already stated. Repetitious

L258 Subheading 3.8  I don't see that this has anything to do with the "Challenges In Archaeal Culturomics" which is the main heading.

L259-260  A winding sentence. Be concise.

Figure 2  This figure is most uninformative.

L267  And we the readers hope that you will reveal wisdom regarding this.

L267-268  This is probably not unique to the archaea.

L276  At this point it would be interesting to take a hard look at the genomics of a sequenced and reasonably well-studied archaea to see what can be gleaned from the annotated genome and what we know to be media that is successful in cultivation of the bug. Knowing the genome, how could one expand the formulations of media for this particular bug? This would help the reader enormously in considering possibilities for media selection and would provide valuable guidance for linking genomics and cultivation.

L282  Already stated.

L290-291  Confusing sentence. Note that "different strategies" is the subject which can be used to isolate...most of which depend on...characteristics of organisms it comprises"??? What is exactly "comprising" these organisms. A murky sentence.

L295-297  repetitious

L304  was achieved

L327  replace team mate with colleagues

L342-346  A long confusing incomplete sentence.

L357-358  highly repetitious

L362-364  This should be in the introduction.

L364-366  This sentence needs to be recast. As it stands it makes little sense.

L367-369  This is repetitious and should be in the intro where you are making the case that there are many problems.

L379  why is it "inadequate"?

L380  "need for proteins from dead organisms"  needs a reference

L380-382  The logic that drives this conclusion is not clear.

L382  What is "in-lab media"??

Figure 4 legend. Details on these methods would be welcome. Also a critical analysis of these same methods. 

L383  If it was only extra vigilance that was required then they would all be cultivated by now. 

L386  Grammar "are the only life forms"

L388  why "external"

L388-389  Some would argue that CO2 and H2 are part of the components of regular media.

L390-391  Far more helpful to the reader would be a critical analysis of each component of "typical" archaea media. Why is it included and what logic drives the concentrations of all these components.

L398  requires "a" particular nutrient?? Please inform us what is this "particular nutrient" .

L402  Grammar  and grow more easily...

L417-418   or what is "it" in this sentence?

L422-423  This is a wonderful thought. It would be much better if the authors fleshed out how this was done. For example, as a rapid method of determining if a particular component enhanced archaeal growth, one could inoculate microtiter plates with a substantial amount of the habitat (eg. 200 µl of water known to contain archaea) and then supplement wells with individual media components and combinations of components. Incubate and test for increases in archaea PCR signal, indicating that component(s) had stimulated archaea growth.

L444 Reflection ??

L445-446  Not a good sentence.  Perhaps here is where you should use "reflect". Lose the comma and it is not the methods and procedures that are to blame. "Current efforts in cultivating the unculturable Archaea do not reflect the importance of the domain in the biosphere."

Comments on the Quality of English Language

There are several aspects of language that need to be addressed by the authors. Basic grammar is abandoned in parts of the paper. Insufficient attention to logical organization of the information is apparent. Imprecise use of words continually trip up the reader. Avoid repeating the same point 4-5 times. My suggestion is to reorganize the paper and then rewrite it. Do not simply regurgitate what is in the literature, but attempt a critical assessment of the current state of cultivation of the archaea and then make clearly stated recommendations for future work.

Author Response

(The authors gave the same response as above.)

Reviewer 4 Report

Comments and Suggestions for Authors

This manuscript presents an incomplete review of the literature concerning culturing of Archaeal taxa. A lot of sections are repetitive, and very few recent (2020 - 2023) studies are mentioned. Several figures are misleading. The conclusions and suggestions are not helpful or well informed.  

The English in this manuscript needs full and comprehensive review by a native speaker, as almost every sentence uses odd phrasing.

Specific comments:

line 38 "opinion" should be "knowledge"

Figure 1 should be removed. It contains misleading categories and erroneous information. Crenarchaeota and Thaumarchaeaota are members of TACK (as the authors mention in line 93) but are shown separately from TACK - why? The cultureable phyla are shown in blue but the legend says the blue colour means unculturable. The habitats colours for soil and hydrothermal vent are too similar, and the habitats given for each phylum/superphylum do not correlate with known archaeal abundances based on 16S gene surveys. 

Providing only two references for cultivable archaeal isolates seems a remarkably incomplete list (line 162), and also the references are very old. What about more recent studies?  

In fact, most of the paper has only few or very old references, and has omitted newer achievements in archaeal culturing approaches. 

Figure 2 needs more explanation - what are the yellow, purple, green and fluoro cells? How does the cycle of arrows below relate to the circle above, when the death phase is shown on the side of the dividable cells?

Line 259 states that data about genomes of cultivable archaea is scarce, unavailable and wrongly annotated. This is not correct at all. The reference of Sun et al 2020 certainly does not support this claim.  And then the rest of the paragraph does not talk about genomes at all!

Figure 3 has many spelling errors.

line 357 "by can culturing them" - what do you mean here? 

The authors claim that culturing halophilic archaea is difficult (line 382). Halophilic archaea are one group of the archaea that are most readily cultured, and for which there are a lot of pure isolates available. Please do more comprehensive literature searches!

Line 451 "On basis of available media in  present review, we suggested basal or general purpose medium for culturing of Archaea." The authors conclude that a general purpose medium should be used for culturing archaea, after just listing out exactly how different the growth requirements for various phyla are - this is an inane conclusion to make, especially as the authors go on in the very next sentence to contradict themselves by calling for formulation of new media according to information from genomic, metagenomic and co-culturing studies. 

Comments on the Quality of English Language

Extensive editing of the English is required.

Author Response

(The authors gave the same response as above.)

Round 2

Reviewer 1 Report

Comments and Suggestions for Authors

Please correct the spelling of the subtitle: lane 123

Author Response

The reviewer's comments were carefully addressed. The file is attached for detailed point-to-point responses. 

Reviewer 3 Report

Comments and Suggestions for Authors

Challenges and Approaches of Culturing the Unculturable Archaea

The manuscript has been significantly changed in response to reviewer's comments. The author's appear to have greatly reduced needless repetition.  Another reviewer comment was reproofing to correct the strained english in some sentences. This has improved but there are still a many problem areas. For example in the Simple Summary the following; "Comparatively, the two domains, eukarya and bacteria, are extensively studied for the taxonomy, diversity, ecology, physiology, and clinical and biotechnological aspects than archaea." The sentence does strain the ear. A more direct construction reads better; "The two domains, eukarya and bacteria, are more extensively studied for the taxonomy, diversity, ecology, physiology and clinical and biotechnological aspects compared to the archaea".

There are also areas where the authors appear to overstate their position. For example in the intro they state: "Based on prior knowledge, isolating bacteria and fungi is comparatively easy, in contrast to the challenges faced in archaeal isolation." In fact there are still many groups of bacteria that are underrepresented in culture collections because of difficulties in isolations.

The figure legend of Fig 1 is a little confusing. It begins with "The red color represents the targeted clades..."  Yet I see no red color in the fig.

The author's state: "Similarly, a phenomenon of plate count anomaly limits the number of retrieved archaea, diversity and actual role from the natural world [56, 59, 60]."  I would argue that it is not the phenomenon of the plate count anomaly that limits the number of retrievable archaea but rather our limited understanding of their optimal growth conditions. the plate count anomaly is simply the phenotype of imperfect growth conditions.

Another sentence that is awkward: "The factors behind using lignin as a source of energy by the specific group of archaea recovered from the marine sediment have multiple copy  number genes coding for lignin-degrading enzymes, its transportation and utilization as  a carbon and energy source."  This example need expansion. It is not clear to the reader from this presentation whether or not the genomic information informed the media formulation or not. If so it should be explicitly stated and referenced. The authors go on to state "The evolutionary mechanisms over time in lignin-rich ex-247 treme habitats...". This begs the question of when lignins first appeared as this would be a marker for the evolution of lignin degradation genes.

Persister cells are commonly present in Bacteria. Be precise, Persister cells are not present in bacteria. They may be present in populations of bacteria.

Another cryptic sentence "Data about genome sequences of pure and cultivable archaea is scarce, which depicts the unavailability of genome sequences for annotation and extraction of proper metabolic pathways [93]."  Given that genomic data on the archaea is scarce, there is little for annotation.  There is repetition within the sentence...."genomic data is scarce...depicts the unavailability of genome sequences". this is very strained. In fact what has led to poorly annotated archaea genomes is that there are far fewer cultivated archaea that have been genetically, physiologically and biochemically investigated - hence fewer known gene functions.

There are more problematic sentences that should be refined. Another reviewer comment noted that this review skimmed along the surface of the literature and provided little critical insight into the problems of cultivation. This has been addressed at some level but the review still seems superficial to me.

Comments on the Quality of English Language

-

Author Response

Detail of the responses to each comment is given in the attached file.

Reviewer 4 Report

Comments and Suggestions for Authors

The text of the review article is now much improved, and I appreciate that a lot of new content and more up-to-date references have been added. 

Muhsin Jamal has been added as an author – could this person’s contributions be included in the Author Contributions statement please (line 613)

Figure 1 still has major issues and it should be removed. Information about the main habitats of certain archaeal types could be added to the main text instead. Current errors in Figure 1 include:

·       the Ammonia-oxidizing archaea  (Marked in the figure as members of Crenarchaeota and as unculturable) are members of phylym Thaumarchaeota (NCBI) / Thermoproteota (GTDB) and some have been cultured on solid media as reported by Klein et al 2022. https://doi.org/10.1093/femsle/fnac029. This paper should also be discussed in section 4 or 5 when you talk about successful alternative culturing techniques later in the review. 

·       The Euryarchaeota (B) contain many archaea that have been cultured (Halophilic archaea, methanogens), so this section should have some green to show that some have been cultured. 

·       The Figure legend says there is red representing targeted clades but there is no red in the figure? 

·       The figure legend talks about A) as methane oxidizing archaea (Methanomicrobia) – Methanomicrobia belong in Euryarchaeota not Thaumarchaeota as section A of the figure is labelled.

Figure 2 and it’s legend has been made much clearer – thank you.

Line 65: The GTDB currently has 21 phyla for the domain Archaea, with a total of 7777 species. https://gtdb.ecogenomic.org/tree?r=d__Archaea

Line 334 – change “Therefore, most archaeal genes have been wrongly annotated or linked with unknown and hypothetical proteins, indicating a genomic lack of information for archaeal cultivation.”  to “Therefore, some archaeal genes may have been wrongly annotated. or linked with unknown and hypothetical proteins, indicating a genomic lack of information for archaeal cultivation.”  You go on to talk about how 30% - 80% of proteins in an archaeal genome can be hypotheticals, so the crossed out text is not needed.   

Please add more discussion of successful uses of co-culturing in Section 4 of the manuscript – eg. two Asgard archaea that were able to be highly enriched in enrichment cultures (Imachi et al 2020; Rodrigues-Oliveira et al 2023 Nature 613:332-339). When a highly enriched culture is obtained then other experiments to characterise the organism can be done (FISH, cell structure microscopy, stable isotope probing) and this can then guide further isolation attempts.

Over all, I think the manuscript has improved a lot from the first version, and with these last set of changes it will be suitable for publication.  Please also do one more English language check – particularly for spelling, tenses and capitalization.

Comments on the Quality of English Language

Please do one more English language check – particularly for spelling, tenses and capitalization.

Author Response

A file is attached for the detail point-to-point response of the comments. 
